# Monotone $k$-Submodular Function Maximization with Size Constraints

**Naoto Ohsaka**
The University of Tokyo
ohsaka@is.s.u-tokyo.ac.jp

**Yuichi Yoshida**
National Institute of Informatics, and
Preferred Infrastructure, Inc.
yyoshida@nii.ac.jp

## Abstract

A $k$-submodular function is a generalization of a submodular function, where the input consists of $k$ disjoint subsets, instead of a single subset, of the domain. Many machine learning problems, including influence maximization with $k$ kinds of topics and sensor placement with $k$ kinds of sensors, can be naturally modeled as the problem of maximizing monotone $k$-submodular functions. In this paper, we give constant-factor approximation algorithms for maximizing monotone $k$-submodular functions subject to several size constraints. The running time of our algorithms are almost linear in the domain size. We experimentally demonstrate that our algorithms outperform baseline algorithms in terms of the solution quality.

## 1 Introduction

The task of selecting a set of items subject to constraints on the size or the cost of the set is versatile in machine learning problems. The objective can be often modeled as maximizing a function with the diminishing return property, where for a finite set $V$, a function $f : 2^V \to \mathbb{R}$ satisfies the *diminishing return property* if

$$f(S \cup \{e\}) - f(S) \geq f(T \cup \{e\}) - f(T)$$

for any $S \subseteq T$ and $e \in V \setminus T$. For example, sensor placement [13, 14], influence maximization in social networks [11], document summarization [15], and feature selection [12] involve objectives satisfying the diminishing return property. It is well known that the diminishing return property is equivalent to submodularity, where a function $f : 2^V \to \mathbb{R}$ is *submodular* if

$$f(S) + f(T) \geq f(S \cap T) + f(S \cup T)$$

holds for any $S, T \subseteq V$. When the objective function is submodular and hence satisfies the diminishing return property, we can find in polynomial time a solution with a provable guarantee on its solution quality even with various constraints [2, 3, 18, 21].

In many practical applications, however, we want to select several disjoint sets of items instead of a single set. To see this, let us describe two examples:

**Influence maximization:** Viral marketing is a cost-effective marketing strategy that promotes products by giving free (or discounted) items to a selected group of highly influential people in the hope that, through the word-of-mouth effects, a large number of product adoptions will occur [4, 19]. Suppose that we have $k$ kinds of items, each having a different topic and thus a different word-of-mouth effect. Then, we want to distribute these items to $B$ people selected from a group $V$ of $n$ people so as to maximize the (expected) number of product adoptions. It is natural to impose a constraint that each person can receive at most one item since giving many free items to one particular person would be unfair.

**Sensor placement:** There are $k$ kinds of sensors for different measures such as temperature, humidity, and illuminance. Suppose that we have $B_i$ many sensors of the $i$-th kind for each

$i \in \{1, 2, \ldots, k\}$, and there is a set $V$ of $n$ locations, each of which can be instrumented with exactly one sensor. Then, we want to allocate those sensors so as to maximize the information gain.

When $k = 1$, these problems can be modeled as maximizing monotone submodular functions [11, 14] and admit polynomial-time $(1 - 1/e)$-approximation [18]. Unfortunately, however, the case of general $k$ cannot be modeled as maximizing submodular functions, and we cannot apply the methods in the literature on maximizing submodular functions [2, 3, 18, 21]. We note that the problem of selecting $k$ disjoint sets can be sometimes modeled as maximizing monotone submodular functions over the extended domain $k \times V$ subject to a partition matroid. Although $(1 - 1/e)$-approximation algorithms are known [3, 5], the running time is around $O(k^8 n^8)$ and is prohibitively slow.

**Our contributions:**  To address the problem of selecting $k$ disjoint sets, we use the fact that the objectives can be often modeled as $k$-submodular functions. Let $(k + 1)^V := \{(X_1, \ldots, X_k) \mid X_i \subseteq V \ \forall i \in \{1, 2, \ldots, k\}, X_i \cap X_j = \emptyset \ \forall i \neq j\}$ be the family of $k$ disjoint sets. Then, a function $f : (k + 1)^V \to \mathbb{R}$ is called $k$-*submodular* [9] if, for any $\boldsymbol{x} = (X_1, \ldots, X_k)$ and $\boldsymbol{y} = (Y_1, \ldots, Y_k)$ in $(k + 1)^V$, we have

$$f(\boldsymbol{x}) + f(\boldsymbol{y}) \geq f(\boldsymbol{x} \sqcup \boldsymbol{y}) + f(\boldsymbol{x} \sqcap \boldsymbol{y})$$

where

$$\boldsymbol{x} \sqcap \boldsymbol{y} := (X_1 \cap Y_1, \ldots, X_k \cap Y_k),$$
$$\boldsymbol{x} \sqcup \boldsymbol{y} := \Big(X_1 \cup Y_1 \setminus \big(\bigcup_{i \neq 1} X_i \cup Y_i\big), \ldots, X_k \cup Y_k \setminus \big(\bigcup_{i \neq k} X_i \cup Y_i\big)\Big).$$

Roughly speaking, $k$-submodularity captures the property that, if we choose exactly one set $X_e \in \{X_1, \ldots, X_k\}$ that an element $e$ can belong to for each $e \in V$, then the resulting function is submodular (see Section 2 for details). When $k = 1$, $k$-submodularity coincides with submodularity.

In this paper, we give approximation algorithms for maximizing non-negative monotone $k$-submodular functions with several constraints on the sizes of the $k$ sets. Here, we say that $f$ is *monotone* if $f(\boldsymbol{x}) \leq f(\boldsymbol{y})$ for any $\boldsymbol{x} = (X_1, \ldots, X_k)$ and $\boldsymbol{y} = (Y_1, \ldots, Y_k)$ with $X_i \subseteq Y_i$ for each $i \in \{1, \ldots, k\}$. Let $n = |V|$ be the size of the domain. For the *total size constraint*, under which the total size of the $k$ sets is bounded by $B \in \mathbb{Z}_+$, we show that a simple greedy algorithm outputs $1/2$-approximation in $O(knB)$ time. The approximation ratio of $1/2$ is asymptotically tight since the lower bound of $\frac{k+1}{2k} + \epsilon$ for any $\epsilon > 0$ is known even when $B = n$ [10]. Combining the random sampling technique [17], we also give a randomized algorithm that outputs $1/2$-approximation with probability at least $1 - \delta$ in $O(kn \log B \log(B/\delta))$ time. Hence, even when $B$ is as large as $n$, the running time is almost linear in $n$. For the *individual size constraint*, under which the size of the $i$-th set is bounded by $B_i \in \mathbb{Z}_+$ for each $i \in \{1, \ldots, k\}$, we give a $1/3$-approximation algorithm with running time $O(knB)$, where $B = \sum_{i=1}^{k} B_i$. We then give a randomized algorithm that outputs $1/3$-approximation with probability at least $1 - \delta$ in $O(k^2 n \log(B/k) \log(B/\delta))$ time.

To show the practicality of our algorithms, we apply them to the influence maximization problem and the sensor placement problem, and we demonstrate that they outperform previous methods based on submodular function maximization and several baseline methods in terms of the solution quality.

**Related work:**  When $k = 2$, $k$-submodularity is called bisubmodularity, and [20] applied bisubmodular functions to machine learning problems. However, their algorithms do not have any approximation guarantee. Huber and Kolmogorov introduced $k$-submodularity as a generalization of submodularity and bisubmodularity [9], and minimizing $k$-submodular functions was successfully used in a computer vision application [8]. Iwata *et al.* [10] gave a $1/2$-approximation algorithm and a $\frac{k}{2k-1}$-approximation algorithm for maximizing non-monotone and monotone $k$-submodular functions, respectively, when there is no constraint.

**Organization:**  The rest of this paper is organized as follows. In Section 2, we review properties of $k$-submodular functions. Sections 3 and 4 are devoted to show $1/2$-approximation algorithms for the total size constraint, and $1/3$-approximation algorithms for the individual size constraint, respectively. We show our experimental results in Section 5. We conclude our paper in Section 6.

---

**Algorithm 1** $k$-Greedy-TS

---

**Input:** a monotone $k$-submodular function $f : (k+1)^V \to \mathbb{R}_+$ and an integer $B \in \mathbb{Z}_+$.
**Output:** a vector $s$ with $|\mathrm{supp}(s)| = B$.

1: $s \leftarrow \mathbf{0}$.
2: **for** $j = 1$ to $B$ **do**
3: $\quad (e, i) \leftarrow \arg\max_{e \in V \setminus \mathrm{supp}(s), i \in [k]} \Delta_{e,i} f(s)$.
4: $\quad s(e) \leftarrow i$.
5: **return** $s$.

---

## 2 Preliminaries

For an integer $k \in \mathbb{N}$, $[k]$ denotes the set $\{1, 2, \ldots, k\}$. We define a partial order $\preceq$ on $(k+1)^V$ so that, for $\boldsymbol{x} = (X_1, \ldots, X_k)$ and $\boldsymbol{y} = (Y_1, \ldots, Y_k)$ in $(k+1)^V$, $\boldsymbol{x} \preceq \boldsymbol{y}$ if $X_i \subseteq Y_i$ for every $i$ with $i \in [k]$. We also define

$$\Delta_{e,i} f(\boldsymbol{x}) = f(X_1, \ldots, X_{i-1}, X_i \cup \{e\}, X_{i+1}, \ldots, X_k) - f(X_1, \ldots, , X_k)$$

for $\boldsymbol{x} \in (k+1)^V$, $e \notin \bigcup_{\ell \in [k]} X_\ell$, and $i \in [k]$, which is the marginal gain when adding $e$ to the $i$-th set of $\boldsymbol{x}$. Then, it is easy to see the monotonicity of $f$ is equivalent to $\Delta_{e,i} f(\boldsymbol{x}) \geq 0$ for any $\boldsymbol{x} = (X_1, \ldots, X_k)$ and $e \notin \bigcup_{\ell \in [k]} X_\ell$ and $i \in [k]$. Also it is not hard to show (see [22] for details) that the $k$-submodularity of $f$ implies the *orthant submodularity*, i.e.,

$$\Delta_{e,i} f(\boldsymbol{x}) \geq \Delta_{e,i} f(\boldsymbol{y})$$

for any $\boldsymbol{x}, \boldsymbol{y} \in (k+1)^V$ with $\boldsymbol{x} \preceq \boldsymbol{y}$, $e \notin \bigcup_{\ell \in [k]} Y_\ell$, and $i \in [k]$, and the *pairwise monotonicity*, i.e.,

$$\Delta_{e,i} f(\boldsymbol{x}) + \Delta_{e,j} f(\boldsymbol{x}) \geq 0$$

for any $\boldsymbol{x} \in (k+1)^V$, $e \notin \bigcup_{\ell \in [k]} X_\ell$, and $i, j \in [k]$ with $i \neq j$. Actually, the converse holds:

**Theorem 2.1** (Ward and Živný [22])**.** *A function $f : (k+1)^V \to \mathbb{R}$ is $k$-submodular if and only if $f$ is orthant submodular and pairwise monotone.*

It is often convenient to identify $(k+1)^V$ with $\{0, 1 \ldots, k\}^V$ to analyze $k$-submodular functions, Namely, we associate $(X_1, \ldots, X_k) \in (k+1)^V$ with $\boldsymbol{x} \in \{0, 1, \ldots, k\}^V$ by $X_i = \{e \in V \mid \boldsymbol{x}(e) = i\}$ for $i \in [k]$. Hence we sometimes abuse notation, and simply write $\boldsymbol{x} = (X_1, \ldots, X_k)$ by regarding a vector $\boldsymbol{x}$ as disjoint $k$ sets of $V$. We define the *support* of $\boldsymbol{x} \in \{0, 1, \ldots, k\}^V$ as $\mathrm{supp}(\boldsymbol{x}) = \{e \in V \mid \boldsymbol{x}(e) \neq 0\}$. Analogously, for $\boldsymbol{x} \in \{0, 1, \ldots, k\}^V$ and $i \in [k]$, we define $\mathrm{supp}_i(\boldsymbol{x}) = \{e \in V \mid \boldsymbol{x}(e) = i\}$. Let $\mathbf{0}$ be the zero vector in $\{0, 1, \ldots, k\}^V$.

## 3 Maximizing $k$-submodular Functions with the Total Size Constraint

In this section, we give a $1/2$-approximation algorithm to the problem of maximizing monotone $k$-submodular functions subject to the total size constraint. Namely, we consider

$$\max f(\boldsymbol{x}) \qquad \text{subject to } |\mathrm{supp}(\boldsymbol{x})| \leq B \text{ and } \boldsymbol{x} \in (k+1)^V,$$

where $f : (k+1)^V \to \mathbb{R}_+$ is monotone $k$-submodular and $B \in \mathbb{Z}_+$ is a non-negative integer.

### 3.1 A greedy algorithm

The first algorithm we propose is a simple greedy algorithm (Algorithm 1). We show the following:

**Theorem 3.1.** *Algorithm 1 outputs a $1/2$-approximate solution by evaluating $f$ $O(knB)$ times, where $n = |V|$.*

The number of evaluations of $f$ is clearly $O(knB)$. Hence in what follows, we focus on analyzing the approximation ratio of Algorithm 1. Our analysis is based on the framework of [10].

Consider the $j$-th iteration of the for loop from Line 2. Let $(e^{(j)}, i^{(j)}) \in V \times [k]$ be the pair greedily chosen in this iteration, and let $\boldsymbol{s}^{(j)}$ be the solution *after* this iteration. We define $\boldsymbol{s}^{(0)} = \mathbf{0}$. Let $\boldsymbol{o}$ be

**Algorithm 2** $k$-Stochastic-Greedy-TS

---

**Input:** a monotone $k$-submodular function $f : (k + 1)^V \to \mathbb{R}_+$, an integer $B \in \mathbb{Z}_+$, and a failure probability $\delta > 0$.

**Output:** a vector $\boldsymbol{s}$ with $|\mathrm{supp}(\boldsymbol{s})| = B$.

1: $\boldsymbol{s} \leftarrow \boldsymbol{0}$.
2: **for** $j = 1$ to $B$ **do**
3: $\quad R \leftarrow$ a random subset of size $\min\{\frac{n-j+1}{B-j+1} \log \frac{B}{\delta}, n\}$ uniformly sampled from $V \setminus \mathrm{supp}(\boldsymbol{s})$.
4: $\quad (e, i) \leftarrow \arg\max_{e \in R, i \in [k]} \Delta_{e,i} f(\boldsymbol{s})$.
5: $\quad \boldsymbol{s}(e) \leftarrow i$.
6: **return** $\boldsymbol{s}$.

---

the optimal solution. We iteratively define $\boldsymbol{o}^{(0)} = \boldsymbol{o}, \boldsymbol{o}^{(1)}, \ldots, \boldsymbol{o}^{(B)}$ as follows. For each $j \in [B]$, let $S^{(j)} = \mathrm{supp}(\boldsymbol{o}^{(j-1)}) \setminus \mathrm{supp}(\boldsymbol{s}^{(j-1)})$. Then, we set $o^{(j)} = e^{(j)}$ if $e^{(j)} \in S^{(j)}$, and set $o^{(j)}$ to be an arbitrary element in $S^{(j)}$ otherwise. Then, we define $\boldsymbol{o}^{(j-1/2)}$ as the resulting vector obtained from $\boldsymbol{o}^{(j-1)}$ by assigning 0 to the $o^{(j)}$-th element, and then define $\boldsymbol{o}^{(j)}$ as the resulting vector obtained from $\boldsymbol{o}^{(j-1/2)}$ by assigning $i^{(j)}$ to the $e^{(j)}$-th element. Note that $\mathrm{supp}(\boldsymbol{o}^{(j)}) = B$ holds for every $j \in \{0, 1, \ldots, B\}$ and $\boldsymbol{o}^{(B)} = \boldsymbol{s}^{(B)} = \boldsymbol{s}$. Moreover, we have $\boldsymbol{s}^{(j-1)} \preceq \boldsymbol{o}^{(j-1/2)}$ for every $j \in [B]$.

*Proof of Theorem 3.1.* We first show that, for each $j \in [B]$,

$$f(\boldsymbol{s}^{(j)}) - f(\boldsymbol{s}^{(j-1)}) \geq f(\boldsymbol{o}^{(j-1)}) - f(\boldsymbol{o}^{(j)}). \tag{1}$$

For each $j \in [B]$, let $y^{(j)} = \Delta_{e^{(j)}, i^{(j)}} f(\boldsymbol{s}^{(j-1)})$, $a^{(j-1/2)} = \Delta_{o^{(j)}, \boldsymbol{o}^{(j-1)}(o^{(j)})} f(\boldsymbol{o}^{(j-1/2)})$, and $a^{(j)} = \Delta_{e^{(j)}, i^{(j)}} f(\boldsymbol{o}^{(j-1/2)})$. Then, note that $f(\boldsymbol{s}^{(j)}) - f(\boldsymbol{s}^{(j-1)}) = y^{(j)}$, and $f(\boldsymbol{o}^{(j-1)}) - f(\boldsymbol{o}^{(j)}) = a^{(j-1/2)} - a^{(j)}$. From the monotonicity of $f$, it suffices to show that $y^{(j)} \geq a^{(j-1/2)}$. Since $e^{(j)}$ and $i^{(j)}$ are chosen greedily, we have $y^{(j)} \geq \Delta_{o^{(j)}, \boldsymbol{o}^{(j-1)}(o^{(j)})} f(\boldsymbol{s}^{(j-1)})$. Since $\boldsymbol{s}^{(j-1)} \preceq \boldsymbol{o}^{(j-1/2)}$, we have $\Delta_{o^{(j)}, \boldsymbol{o}^{(j-1)}(o^{(j)})} f(\boldsymbol{s}^{(j-1)}) \geq a^{(j-1/2)}$ from the orthant submodularity. Combining these two inequalities, we establish (1).

Then, we have

$$f(\boldsymbol{o}) - f(\boldsymbol{s}) = \sum_{j=1}^{B}(f(\boldsymbol{o}^{(j-1)}) - f(\boldsymbol{o}^{(j)})) \leq \sum_{j=1}^{B}(f(\boldsymbol{s}^{(j)}) - f(\boldsymbol{s}^{(j-1)})) = f(\boldsymbol{s}) - f(\boldsymbol{0}) \leq f(\boldsymbol{s}),$$

which implies $f(\boldsymbol{s}) \geq f(\boldsymbol{o})/2$. $\qquad\qquad\qquad\qquad\qquad\qquad\qquad\qquad\qquad\qquad\qquad\qquad \square$

## 3.2 An almost linear-time algorithm by random sampling

In this section, we improve the number of evaluations of $f$ from $O(knB)$ to $O(kn \log B \log \frac{B}{\delta})$, where $\delta > 0$ is a failure probability.

Our algorithm is shown in Algorithm 2. The main difference from Algorithm 1 is that we sample a sufficiently large subset $R$ of $V$, and then greedily assign a value only looking at elements in $R$.

We reuse notations $e^{(j)}, i^{(j)}, S^{(j)}$ and $\boldsymbol{s}^{(j)}$ from Section 3.1, and let $R^{(j)}$ be $R$ in the $j$-th iteration. We iteratively define $\boldsymbol{o}^{(0)} = \boldsymbol{o}, \boldsymbol{o}^{(1)}, \ldots, \boldsymbol{o}^{(B)}$ as follows. If $R^{(j)} \cap S^{(j)}$ is empty, then we regard that the algorithm failed. Suppose $R^{(j)} \cap S^{(j)}$ is non-empty. Then, we set $o^{(j)} = e^{(j)}$ if $e^{(j)} \in R^{(j)} \cap S^{(j)}$, and set $o^{(j)}$ to be an arbitrary element in $R^{(j)} \cap S^{(j)}$ otherwise. Finally, we define $\boldsymbol{o}^{(j-1/2)}$ and $\boldsymbol{o}^{(j)}$ as in Section 3.1 using $\boldsymbol{o}^{(j-1)}$, $o^{(j)}$, and $e^{(j)}$.

If the algorithm does not fail and $\boldsymbol{o}^{(1)}, \ldots, \boldsymbol{o}^{(B)}$ are well defined, or in other words, if $R^{(j)} \cap S^{(j)}$ is not empty for every $j \in [B]$, then the rest of the analysis is completely the same as in Section 3.1, and we achieve an approximation ratio of $1/2$. Hence, it suffices to show that $\boldsymbol{o}^{(1)}, \ldots, \boldsymbol{o}^{(B)}$ are well defined with a high probability.

**Lemma 3.2.** *With probability at least $1 - \delta$, we have $R^{(j)} \cap S^{(j)} \neq \emptyset$ for every $j \in [B]$.*

**Algorithm 3** $k$-Greedy-IS
___
**Input:** a monotone $k$-submodular function $f : (k+1)^V \to \mathbb{R}_+$ and integers $B_1, \ldots, B_k \in \mathbb{Z}_+$.
**Output:** a vector $s$ with $|\mathrm{supp}_i(s)| = B_i$ for each $i \in [k]$.
1: $s \leftarrow \mathbf{0}$ and $B \leftarrow \sum_{i \in [k]} B_i$.
2: **for** $j = 1$ to $B$ **do**
3: $\quad I \leftarrow \{i \in [k] \mid \mathrm{supp}_i(s) < B_i\}$.
4: $\quad (e, i) \leftarrow \arg\max_{e \in V \setminus \mathrm{supp}(s), i \in I} \Delta_{e,i} f(s)$.
5: $\quad s(e) \leftarrow i$.
6: **return** $s$.
___

*Proof.* Fix $j \in [B]$. If $|R^{(j)}| = n$, then we clealy have $\Pr[R^{(j)} \cap S^{(j)} = \emptyset] = 0$. Otherwise we have

$$\Pr[R^{(j)} \cap S^{(j)} = \emptyset] = \left( 1 - \frac{|S^{(j)}|}{|V \setminus \mathrm{supp}(s^{(j-1)})|} \right)^{|R^{(j)}|} \leq e^{-\frac{B-j+1}{n-j+1} \frac{n-j+1}{B-j+1} \log \frac{B}{\delta}} = \frac{\delta}{B}.$$

By the union bound over $j \in [B]$, the lemma follows. $\qquad\square$

**Theorem 3.3.** *Algorithm 2 outputs a $1/2$-approximate solution with probability at least $1 - \delta$ by evaluating $f$ at most $O(k(n - B) \log B \log \frac{B}{\delta})$ times.*

*Proof.* By Lemma 3.2 and the analysis in Section 3.1, Algorithm 2 outputs a $1/2$-approximate solution with probability at least $1 - \delta$.

The number of evaluations of $f$ is at most

$$k \sum_{j \in [B]} \frac{n - j + 1}{B - j + 1} \log \frac{B}{\delta} = k \sum_{j \in [B]} \frac{n - B + j}{j} \log \frac{B}{\delta} = O\left( kn \log B \log \frac{B}{\delta} \right). \qquad\square$$

# 4 Maximizing $k$-submodular Functions with the Individual Size Constraint

In this section, we consider the problem of maximizing monotone $k$-submodular functions subject to the individual size constraint. Namely, we consider

$$\max f(x) \qquad \text{subject to } |\mathrm{supp}_i(x)| \leq B_i \ \forall i \in [k] \text{ and } x \in (k+1)^V,$$

where $f : (k+1)^V \to \mathbb{R}_+$ is monotone $k$-submodular, and $B_1, \ldots, B_k \in \mathbb{Z}_+$ are non-negative integers.

## 4.1 A greedy algorithm

We first consider a simple greedy algorithm described in Algorithm 3. We show the following:

**Theorem 4.1.** *Algorithm 3 outputs a $1/3$-approximate solution by evaluating $f$ at most $O(knB)$ times.*

It is clear that the number of evaluations of $f$ is $O(knB)$. The analysis of the approximation ratio is given in Appendix A.

## 4.2 An almost linear-time algorithm by random sampling

We next improve the number of evaluations of $f$ from $O(knB)$ to $O\left( k^2 n \log \frac{B}{k} \log \frac{B}{\delta} \right)$. Our algorithm is given in Algorithm 4. In Appendix A, we show the following.

**Theorem 4.2.** *Algorithm 4 outputs a $1/3$-approximate solution with probability at least $1 - \delta$ by evaluating $f$ at most $O\left( k^2 n \log \frac{B}{k} \log \frac{B}{\delta} \right)$ times.*

**Algorithm 4** $k$-Stochastic-Greedy-IS

---

**Input:** a monotone $k$-submodular function $f : (k+1)^V \to \mathbb{R}_+$, integers $B_1, \ldots, B_k \in \mathbb{Z}_+$, and a failure probability $\delta > 0$.

**Output:** a vector $\boldsymbol{s}$ with $|\text{supp}_i(\boldsymbol{s})| = B_i$ for each $i \in [k]$.

  1: $\boldsymbol{s} \leftarrow \boldsymbol{0}$ and $B \leftarrow \sum_{i \in [k]} B_i$.
  2: **for** $j = 1$ to $B$ **do**
  3:     $I \leftarrow \{i \in [k] \mid \text{supp}_i(\boldsymbol{s}) < B_i\}$ and $R \leftarrow \emptyset$.
  4:     **loop**
  5:         Add a random element in $V \setminus (\text{supp}(\boldsymbol{s}) \cup R)$ to $R$.
  6:         $(e, i) \leftarrow \arg\max_{e \in R, i \in I} \Delta_{e,i} f(\boldsymbol{s})$.
  7:         **if** $|R| \geq \min\{\frac{n - |\text{supp}_i(\boldsymbol{s})|}{B_i - |\text{supp}_i(\boldsymbol{s})|} \log \frac{B}{\delta}, n\}$ **then**
  8:             $\boldsymbol{s}(e) \leftarrow i$.
  9:             **break** the loop.
10: **return** $\boldsymbol{s}$

---

## 5 Experiments

In this section, we experimentally demonstrate that our algorithms outperform baseline algorithms and our almost linear-time algorithms significantly improve efficiency in practice. We conducted experiments on a Linux server with Intel Xeon E5-2690 (2.90 GHz) and 264GB of main memory. We implemented all algorithms in C++. We measured the computational cost in terms of the number of function evaluations so that we can compare the efficiency of different methods independently from concrete implementations.

### 5.1 Influence maximization with $k$ topics under the total size constraint

We first apply our algorithms to the problem of maximizing the spread of influence on several topics. First we describe our information diffusion model, called the *$k$-topic independent cascade* ($k$-IC) model, which generalizes the *independent cascade* model [6, 7]. In the $k$-IC model, there are $k$ kinds of items, each having a different topic, and thus $k$ kinds of rumors independently spread through a social network. Let $G = (V, E)$ be a social network with an edge probability $p_{u,v}^i$ for each edge $(u, v) \in E$, representing the strength of influence from $u$ to $v$ on the $i$-th topic. Given a seed $\boldsymbol{s} \in (k+1)^V$, for each $i \in [k]$, the diffusion process of the rumor about the $i$-th topic starts by activating vertices in $\text{supp}_i(\boldsymbol{s})$, independently from other topics. Then the process unfolds in discrete steps according to the following randomizes rule: When a vertex $u$ becomes active in the step $t$ for the first time, it is given a single chance to activate each current inactive vertex $v$. It succeeds with probability $p_{u,v}^i$. If $u$ succeeds, then $v$ becomes active in the step $t + 1$. Whether or not $u$ succeeds, it cannot make any further attempt to activate $v$ in subsequent steps. The process runs until no more activation is possible.

The *influence spread* $\sigma : (k+1)^V \to \mathbb{R}_+$ in the $k$-IC model is defined as the expected total number of vertices who eventually become active in one of the $k$ diffusion processes given a seed $\boldsymbol{s}$, namely, $\sigma(\boldsymbol{s}) = \mathbf{E}\left[|\bigcup_{i \in [k]} A_i(\text{supp}_i(\boldsymbol{s}))|\right]$, where $A_i(\text{supp}_i(\boldsymbol{s}))$ is a random variable representing the set of activated vertices in the diffusion process of the $i$-th topic. Given a directed graph $G = (V, E)$, edge probabilities $p_{u,v}^i$ $((u, v) \in E, i \in [k])$, and a budget $B$, the problem is to select a seed $\boldsymbol{s} \in (k+1)^V$ that maximizes $\sigma(\boldsymbol{s})$ subject to $|\text{supp}(\boldsymbol{s})| \leq B$. It is easy to see that the influence spread function $\sigma$ is monotone $k$-submodular (see Appendix B for the proof).

**Experimental settings:** We use a publicly available real-world dataset of a social news website Digg.[1] This dataset consists of a directed graph where each vertex represents a user and each edge represents the friendship between a pair of users, and a log of user votes for stories. We set the number of topics $k$ to be 10, and estimated edge probabilities on each topic from the log using the method of [1]. We set the value of $B$ to $5, 10, \ldots, 100$ and compared the following algorithms:

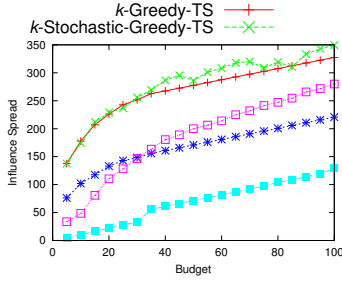

Figure 1: Comparison of influence spreads.

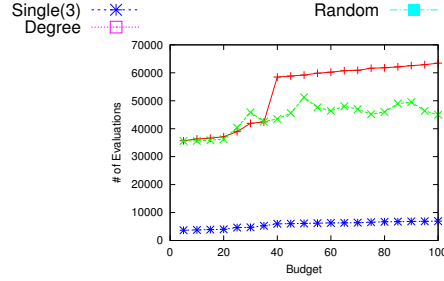

Figure 2: The number of influence estimations.

- $k$-Greedy-TS (Algorithm 1).
- $k$-Stochastic-Greedy-TS (Algorithm 2). We chose $\delta = 0.1$.
- Single($i$): Greedily choose $B$ vertices only considering the $i$-th topic and assign them items of the $i$-th topic.
- Degree: Choose $B$ vertices in decreasing order of degrees and assign them items of random topics.
- Random: Randomly choose $B$ vertices and assign them items of random topics.

For the first three algorithms, we implemented the lazy evaluation technique [16] for efficiency. For $k$-Greedy-TS, we maintain an upper bound on the gain of inserting each pair $(e, i)$ to apply the lazy evaluation technique directly. For $k$-Stochastic-Greedy-TS, we maintain an upper bound on the gain for each pair $(e, i)$, and we pick up a pair in $R$ with the largest gain for each iteration. During the process of the algorithms, the influence spread was approximated by simulating the diffusion process 100 times. When the algorithms terminate, we simulated the diffusion process 10,000 times to obtain sufficiently accurate estimates of the influence spread.

**Results:**  Figure 1 shows the influence spread achieved by each algorithm. We only show Single(3) among Single($i$) strategies since its influence spread is the largest. $k$-Greedy-TS and $k$-Stochastic-Greedy-TS clearly outperform the other methods owing to their theoretical guarantee on the solution quality. Note that our two methods simulated the diffusion process 100 times to choose a seed set, which is relatively small, because of the high computation cost. Consequently, the approximate value of the influence spread has a relatively high variance, and this might have caused the greedy method to choose seeds with small influence spreads. Remark that Single(3) works worse than Degree for $B$ larger than 35, which means that focusing on a single topic may significantly degrade the influence spread. Random shows a poor performance as expected.

Figure 2 reports the number of influence estimations of greedy algorithms. We note that $k$-Stochastic-Greedy-TS outperforms $k$-Greedy-TS, which implies that the random sampling technique is effective even when combined with the lazy evaluation technique. The number of evaluations of $k$-Greedy-TS drastically increases when $B$ is around 40 since we run out of influential vertices and we need to reevaluate the remaining vertices. Indeed, the slope of $k$-Greedy-TS after $B = 40$ is almost constant in Figure 1, which indicates that the remaining vertices have a similar influence. Single(3) is faster than our algorithms since it only considers a single topic.

### 5.2  Sensor placement with $k$ kinds of measures under the individual size constraint

Next we apply our algorithms for maximizing $k$-submodular functions with the individual size constraint to the sensor placement problem that allows multiple kinds of sensors. In this problem, we want to determine the placement of multiple kinds of sensors that most effectively reduces the expected uncertainty. We need several notions to describe our model. Let $\Omega = \{X_1, X_2, \ldots, X_n\}$ be a set of discrete random variables. The *entropy* of a subset $\mathcal{S}$ of $\Omega$ is defined as $H(\mathcal{S}) = -\sum_{\boldsymbol{s} \in \mathrm{dom}\, \mathcal{S}} \Pr[\boldsymbol{s}] \log \Pr[\boldsymbol{s}]$. The *conditional entropy* of $\Omega$ having observed $\mathcal{S}$ is $H(\Omega \mid \mathcal{S}) := H(\Omega) - H(\mathcal{S})$. Hence, in order to reduce the uncertainty of $\Omega$, we want to find a set $\mathcal{S}$ of as a large entropy as possible.

Now we formalize the sensor placement problem. There are $k$ kinds of sensors for different measures. Suppose that we want to allocate $B_i$ many sensors of the $i$-th kind for each $i \in [k]$, and there

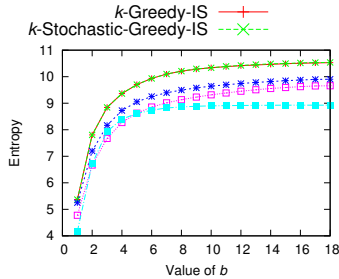

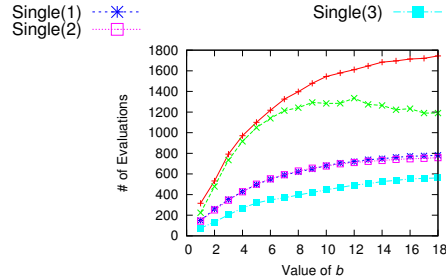

Figure 3: Comparison of entropy.    Figure 4: The number of entropy evaluations.

are set $V$ of $n$ locations, each of which can be instrumented with exactly one sensor. Let $X_e^i$ be the random variable representing the observation collected from a sensor of the $i$-th kind if it is installed at the $e$-th location, and let $\Omega = \{X_e^i\}_{i\in[k],e\in V}$. Then, the problem is to select $\boldsymbol{x} \in (k+1)^V$ that maximizes $f(\boldsymbol{x}) = H\left(\bigcup_{e\in\mathrm{supp}(\boldsymbol{x})}\{X_e^{\boldsymbol{x}(e)}\}\right)$ subject to $|\mathrm{supp}_i(\boldsymbol{x})| \leq B_i$ for each $i \in [k]$. It is easy to see that $f$ is monotone $k$-submodular (see Appendix B for details).

**Experimental settings:**    We use the publicly available Intel Lab dataset.[2] This dataset contains a log of approximately 2.3 million readings collected from 54 sensors deployed in the Intel Berkeley research lab between February 28th and April 5th, 2004. We extracted temperature, humidity, and light values from each reading and discretized these values into several bins of 2 degrees Celsius each, 5 points each, and 100 luxes each, respectively. Hence there are $k = 3$ kinds of sensors to be allocated to $n = 54$ locations. Budgets for sensors measuring temperature, humidity, and light are denoted by $B_1$, $B_2$, and $B_3$. We set $B_1 = B_2 = B_3 = b$, where $b$ is a parameter varying from 1 to 18. We compare the following algorithms:

- $k$-Greedy-IS (Algorithm 3).
- $k$-Stochastic-Greedy-IS (Algorithm 4). We chose $\delta = 0.1$.
- Single($i$): Allocate sensors of the $i$-th kind to greedily chosen $\sum_j B_j$ places.

We again implemented these algorithms with the lazy evaluation technique in a similar way to the previous experiment. Also note that Single($i$) strategies do not satisfy the individual size constraint.

**Results:**    Figure 3 shows the entropy achieved by each algorithm. $k$-Greedy-IS and $k$-Stochastic-Greedy-IS clearly outperform Single($i$) strategies. The maximum gap of entropies achieved by $k$-Greedy-IS and $k$-Stochastic-Greedy-IS is only 0.18%.

Figure 4 shows the number of entropy evaluations of each algorithm. We observe that $k$-Stochastic-Greedy-IS outperforms $k$-Greedy-IS. Especially when $b = 18$, the number of entropy evaluations is reduced by 31%. Single($i$) strategies are faster because they only consider sensors of a fixed kind.

## 6 Conclusions

Motivated by real-world applications, we proposed approximation algorithms for maximizing monotone $k$-submodular functions. Our algorithms run in almost linear time and achieve the approximation ratio of $1/2$ for the total size constraint and $1/3$ for the individual size constraint. We empirically demonstrated that our algorithms outperform baseline methods for maximizing submodular functions in terms of the solution quality. Improving the approximation ratio of $1/3$ for the individual size constraint or showing it tight is an interesting open problem.

## Acknowledgments

Y. Y. is supported by JSPS Grant-in-Aid for Young Scientists (B) (No. 26730009), MEXT Grant-in-Aid for Scientific Research on Innovative Areas (24106003), and JST, ERATO, Kawarabayashi Large Graph Project. N. O. is supported by JST, ERATO, Kawarabayashi Large Graph Project.

## Footnotes

[1] http://www.isi.edu/~lerman/downloads/digg2009.html

[2]http://db.csail.mit.edu/labdata/labdata.html

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
