[Supplementary Material]

# A  Proofs from Section 4

## A.1  Proof of Theorem 4.1

We reuse notations $e^{(j)}$, $i^{(j)}$, and $\boldsymbol{s}^{(j)}$ from Section 3. We iteratively define $\boldsymbol{o}^{(0)} = \boldsymbol{o}, \boldsymbol{o}^{(1)}, \ldots, \boldsymbol{o}^{(B)}$ as follows. For each $j \in [B]$, we define $S_i^{(j)} = \mathrm{supp}_i(\boldsymbol{o}^{(j-1)}) \setminus \mathrm{supp}_i(\boldsymbol{s}^{(j-1)})$. We have two cases to consider:

- Suppose that $e^{(j)} \in S_{i'}^{(j)}$ for some $i' \neq i^{(j)}$. In this case, let $o^{(j)}$ be an arbitrary element in $S_{i^{(j)}}^{(j)}$. Then, we define $\boldsymbol{o}^{(j-1/2)}$ as the resulting vector obtained from $\boldsymbol{o}^{(j-1)}$ by assigning $0$ to the $e^{(j)}$-th element and the $o^{(j)}$-th element, and then define $\boldsymbol{o}^{(j)}$ as the resulting vector obtained from $\boldsymbol{o}^{(j-1/2)}$ by assigning $i^{(j)}$ to the $e^{(j)}$-th element and $i'$ to the $o^{(j)}$-th element.
- Suppose that $e^{(j)} \notin S_{i'}^{(j)}$ for any $i' \neq i^{(j)}$. In this case, we set $o^{(j)} = e^{(j)}$ if $e^{(j)} \in S_{i^{(j)}}^{(j)}$, and we set $o^{(j)}$ to be an arbitrary element in $S_{i^{(j)}}^{(j)}$ otherwise. Then, we define $\boldsymbol{o}^{(j-1/2)}$ as the resulting vector obtained from $\boldsymbol{o}^{(j-1)}$ by assigning $0$ to the $o^{(j)}$-th element, and then define $\boldsymbol{o}^{(j)}$ as the resulting vector obtained from $\boldsymbol{o}^{(j-1/2)}$ by assigning $i^{(j)}$ to the $e^{(j)}$-th element.

Note that $|\mathrm{supp}_i(\boldsymbol{o}^{(j)})| = B_i$ holds for every $i \in [k]$ and $j \in \{0, 1, \ldots, B\}$, and $\boldsymbol{o}^{(B)} = \boldsymbol{s}^{(B)} = \boldsymbol{s}$. Moreover, we have $\boldsymbol{s}^{(j-1)} \preceq \boldsymbol{o}^{(j-1/2)}$ for every $j \in [B]$.

*Proof of Theorem 4.1.* We first show that, for each $j \in [B]$,

$$2(f(\boldsymbol{s}^{(j)}) - f(\boldsymbol{s}^{(j-1)})) \geq f(\boldsymbol{o}^{(j-1)}) - f(\boldsymbol{o}^{(j)}). \tag{2}$$

For each $j \in [B]$, let $y^{(j)} = \Delta_{e^{(j)}, i^{(j)}} f(\boldsymbol{s}^{(j-1)})$. We first note that $f(\boldsymbol{s}^{(j)}) - f(\boldsymbol{s}^{(j-1)}) = y^{(j)}$.

We consider the following two cases:

- Suppose that $e^{(j)} \in S_{i'}^{(j)}$ for some $i' \neq i^{(j)}$. Let $a^{(j-1/2)} = \Delta_{o^{(j)}, i^{(j)}} f(\boldsymbol{o}^{(j-1/2)})$, $a^{(j)} = \Delta_{e^{(j)}, i^{(j)}} f(\boldsymbol{o}^{(j-1/2)})$, $b^{(j-1/2)} = \Delta_{e^{(j)}, i'} f(\boldsymbol{o}^{(j-1/2)})$, and $b^{(j)} = \Delta_{o^{(j)}, i'} f(\boldsymbol{o}^{(j-1/2)})$. Note that $f(\boldsymbol{o}^{(j-1)}) - f(\boldsymbol{o}^{(j)}) = a^{(j-1/2)} - a^{(j)} + b^{(j-1/2)} - b^{(j)}$. From the monotonicity of $f$, it suffices to show that $2y^{(j)} \geq a^{(j-1/2)} + b^{(j-1/2)}$. Since $e^{(j)}$ and $i^{(j)}$ are chosen greedily, we have $y^{(j)} \geq \Delta_{o^{(j)}, i^{(j)}} f(\boldsymbol{s}^{(j-1)})$ and $y^{(j)} \geq \Delta_{e^{(j)}, i'} f(\boldsymbol{s}^{(j-1)})$. Also, since $\boldsymbol{s}^{(j-1)} \preceq \boldsymbol{o}^{(j-1/2)}$, we have $\Delta_{o^{(j)}, i^{(j)}} f(\boldsymbol{s}^{(j-1)}) \geq a^{(j-1/2)}$ and $\Delta_{e^{(j)}, i'} f(\boldsymbol{s}^{(j-1)}) \geq b^{(j-1/2)}$ from the orthant submodularity. Combining these inequalities, we get (2).
- Suppose that $e^{(j)} \notin S_{i'}^{(j)}$ for any $i' \neq i^{(j)}$. Let $a^{(j-1/2)} = \Delta_{o^{(j)}, i^{(j)}} f(\boldsymbol{o}^{(j-1/2)})$, and $a^{(j)} = \Delta_{e^{(j)}, i^{(j)}} f(\boldsymbol{o}^{(j-1/2)})$ Note that $f(\boldsymbol{o}^{(j-1)}) - f(\boldsymbol{o}^{(j)}) = a^{(j-1/2)} - a^{(j)}$. From the monotonicity of $f$, it suffices to show that $2y^{(j)} \geq a^{(j-1/2)}$. Since $e^{(j)}$ and $i^{(j)}$ are chosen greedily, we have $y^{(j)} \geq \Delta_{o^{(j)}, i^{(j)}} f(\boldsymbol{s}^{(j-1)})$. Also, since $\boldsymbol{s}^{(j-1)} \preceq \boldsymbol{o}^{(j-1/2)}$, we have $\Delta_{o^{(j)}, i^{(j)}} f(\boldsymbol{s}^{(j-1)}) \geq a^{(j-1/2)}$ from the orthant submodularity. Combining these inequalities, we get (2).

Then, we have

$$f(\boldsymbol{o}) - f(\boldsymbol{s}) = \sum_{j=1}^{B}(f(\boldsymbol{o}^{(j-1)}) - f(\boldsymbol{o}^{(j)})) \leq \sum_{j=1}^{B} 2(f(\boldsymbol{s}^{(j)}) - f(\boldsymbol{s}^{(j-1)})) = 2(f(\boldsymbol{s}) - f(\boldsymbol{0})) \leq 2f(\boldsymbol{s}).$$

Hence, we have $f(\boldsymbol{s}) \geq f(\boldsymbol{o})/3$. □

## A.2  Proof of Theorem 4.2

We reuse notations $e^{(j)}$, $i^{(j)}$, $S_i^{(j)}$, and $\boldsymbol{s}^{(j)}$. Let $R^{(j)}$ be the set of elements sampled in the $j$-th iteration. We iteratively define $\boldsymbol{o}^{(0)} = \boldsymbol{o}, \boldsymbol{o}^{(1)}, \ldots, \boldsymbol{o}^{(B)}$ as follows. If $R^{(j)} \cap S_{i^{(j)}}^{(j)}$ is empty, we regard that the algorithm failed. Otherwise, we have two cases to consider:

- Suppose that $e^{(j)} \in S_{i'}^{(j)}$ for some $i' \neq i(j)$. In this case, let $o^{(j)}$ be an arbitrary element in $R^{(j)} \cap S_{i(j)}^{(j)}$. Then, we define $\boldsymbol{o}^{(j-1/2)}$ and $\boldsymbol{o}^{(j)}$ as in Section 4.1.
- Suppose that $e^{(j)} \notin S_{i'}^{(j)}$ for any $i' \neq i(j)$. In this case, we set $o^{(j)} = e^{(j)}$ if $e^{(j)} \in S_{i(j)}^{(j)}$ (and hence in $R^{(j)} \cap S_{i(j)}^{(j)}$), and we set $o^{(j)}$ to be an arbitrary element in $R^{(j)} \cap S_{i(j)}^{(j)}$ otherwise. Then, we define $\boldsymbol{o}^{(j-1/2)}$ and $\boldsymbol{o}^{(j)}$ as in Section 4.1.

If $\boldsymbol{o}^{(1)}, \ldots, \boldsymbol{o}^{(B)}$ are well defined, or in other words, if $R^{(j)} \cap S_{i(j)}^{(j)}$ is not empty for each $j \in [B]$, then the rest of the analysis is completely the same as in Section 4.1, and we achieve an approximation ratio of $1/3$. Hence, it suffices to show that $\boldsymbol{o}^{(1)}, \ldots, \boldsymbol{o}^{(B)}$ are well defined with a high probability.

**Lemma A.1.** *With probability at least* $1 - \delta$, *we have* $R^{(j)} \cap S_{i(j)}^{(j)}$ *is not empty for every* $j \in [B]$.

*Proof.* Fix $j \in [B]$. If $|R^{(j)}| = n$, then we clearly have $\Pr[R^{(j)} \cap S_{i(j)}^{(j)} \neq \emptyset] = 0$. Otherwise we have

$$\Pr[R^{(j)} \cap S_{i(j)}^{(j)} \neq \emptyset] = \left(1 - \frac{|S_{i(j)}^{(j)}|}{|V \setminus \mathrm{supp}_{i(j)}(\boldsymbol{s}^{(j-1)})|}\right)^{|R^{(j)}|} = \left(1 - \frac{B_{i(j)} - |\mathrm{supp}_{i(j)}(\boldsymbol{s}^{(j-1)})|}{n - |\mathrm{supp}_{i(j)}(\boldsymbol{s}^{(j-1)})|}\right)^{|R^{(j)}|}$$

$$\leq \exp\left(\frac{B_{i(j)} - |\mathrm{supp}_{i(j)}(\boldsymbol{s}^{(j-1)})|}{n - |\mathrm{supp}_{i(j)}(\boldsymbol{s}^{(j-1)})|} \frac{n - |\mathrm{supp}_{i(j)}(\boldsymbol{s}^{(j-1)})|}{B_{i(j)} - |\mathrm{supp}_{i(j)}(\boldsymbol{s}^{(j-1)})|} \log \frac{B}{\delta}\right) = \frac{\delta}{B}.$$

By the union bound over $j \in [B]$, the lemma follows. $\qquad\square$

*Proof of Theorem 4.2.* By Lemma A.1 and the previous analysis in Section 4.1, we have that Algorithm 4 outputs a $1/3$-approximate solution with probability at least $1 - \delta$.

The number of evaluations of $f$ is at most

$$k \sum_{j \in [B]} \frac{n - |\mathrm{supp}_{i(j)}(\boldsymbol{s}^{(j-1)})|}{B_{i(j)} - |\mathrm{supp}_{i(j-1)}(\boldsymbol{s}^{(j)})|} \log \frac{B}{\delta} \leq k \sum_{i \in [k]} \sum_{j \in [B_i]} \frac{n - j + 1}{B_i - j + 1} \log \frac{B}{\delta}$$

$$= k \sum_{i \in [k]} \sum_{j \in [B_i]} \frac{n - B_i + j}{j} \log \frac{B}{\delta} = O\left(k \sum_{i \in [k]} (B_i + (n - B_i) \log B_i) \log \frac{B}{\delta}\right)$$

$$= O\left(kn \log \frac{B}{\delta} \cdot \sum_{i \in [k]} \log B_i\right) = O\left(kn \log \frac{B}{\delta} \cdot k \log \frac{\sum_{i \in [k]} B_i}{k}\right) = O\left(k^2 n \log \frac{B}{k} \log \frac{B}{\delta}\right),$$

where we used the AM-GM inequality in the last line. $\qquad\square$

# B   Proofs from Section 5

## B.1   $k$-submodularity of the influence maximization problem

In this section, we show that the function $\sigma : (k+1)^V \to \mathbb{R}_+$ used in the influence maximization problem is monotone $k$-submodular. In order to show the $k$-submodularity of $\sigma$, it suffices to show that $\sigma$ is orthant submodular by Theorem 2.1. Pairwise monotonicity is obvious since $\sigma$ is monotone.

To show the orthant submodularity of $f$, we first describe a convenient way of handling the diffusion process. Fix topic $i$. Then for each edge $(u, v) \in E$, we preserve it with probability $p_{u,v}^i$ and discard it with the remaining probability. Let $G^i$ be the directed graph consisting of the preserved edges. Given a seed $\boldsymbol{s} \in (k+1)^V$, the set of vertices reachable from $\mathrm{supp}_i(\boldsymbol{s})$ in $G^i$ corresponds to the set $A_i(\mathrm{supp}_i(\boldsymbol{s}))$. Recall that $A_i(\mathrm{supp}_i(\boldsymbol{s}))$ is a random variable. Kempe *et al.* [11] showed that the function $\mathbf{E}[|A_i(\cdot)|]$ is submodular.

Fix $\boldsymbol{x} = (X_1, \ldots, X_k)$ and $\boldsymbol{y} = (Y_1, \ldots, Y_k)$ with $\boldsymbol{x} \preceq \boldsymbol{y}$, $e \notin \bigcup_{\ell \in [k]} Y_\ell$ and $i \in [k]$. We want to show that $\Delta_{e,i} f(\boldsymbol{x}) \geq \Delta_{e,i} f(\boldsymbol{y})$. Note that

$$\Delta_{e,i} f(\boldsymbol{x}) - \Delta_{e,i} f(\boldsymbol{y}) = \mathbf{E}\Big[\big|A_i(X_i \cup \{e\}) \cup \bigcup_{j \neq i} A_j(X_j)\big| - \big|A_i(X_i) \cup \bigcup_{j \neq i} A_j(X_j)\big|\Big]$$
$$- \mathbf{E}\Big[\big|A_i(Y_i \cup \{e\}) \cup \bigcup_{j \neq i} A_j(Y_j)\big| - \big|A_i(Y_i) \cup \bigcup_{j \neq i} A_j(Y_j)\big|\Big]. \quad (3)$$

Let $S = \bigcup_{j \neq i} A_j(X_j)$ and $T = \bigcup_{j \neq i} A_j(Y_j)$. Then,

$$(3) = \mathbf{E}\Big[\big|(A_i(X_i \cup \{e\}) \setminus A_i(X_i)) \setminus S\big|\Big] - \mathbf{E}\Big[\big|(A_i(Y_i \cup \{e\}) \setminus A_i(Y_i)) \setminus T\big|\Big]. \quad (4)$$

Since $S \subseteq T$ for every fixed $G^j$ for $j \neq i$, we have

$$(4) \geq \mathbf{E}\Big[\big|A_i(X_i \cup \{e\}) \setminus A_i(X_i)\big|\Big] - \mathbf{E}\Big[\big|A_i(Y_i \cup \{e\}) \setminus A_i(Y_i)\big|\Big] \geq 0.$$

The last inequality holds from the submodularity of $A_i(\cdot)$.

## B.2 $k$-submodularity of the sensor placement problem

Recall that $\Omega = \{X_e^i\}_{i \in [k], e \in V}$, where $X_e^i$ represents the observation collecting from a sensor of the $i$-th kind at the $e$-th location and $f : (k+1)^V \to \mathbb{R}_+$ was defined as $f(\boldsymbol{y}) = H(\bigcup_{e \in \mathrm{supp}(\boldsymbol{x})} \{X_e^{\boldsymbol{x}(e)}\})$, where $H$ is the entropy function. It is well known that $H$ is monotone submodular. In order to show that $f : (k+1)^V \to \mathbb{R}_+$ is a $k$-submodular function, it suffices to show its pairwise monotonicity and orthant submodularity by Theorem 2.1.

We first show that $f$ is monotone, which particularly implies that $f$ is pairwise monotone. Let $\boldsymbol{y} = (Y_1, \ldots, Y_k) \in (k+1)^V$. Then, we can associate $\boldsymbol{y}$ with a set $\mathcal{S}_{\boldsymbol{y}} = \{X_e^i \mid i \in [k], e \in Y_i\}$. Then for any $i \in [k]$, and $e \in V \setminus \bigcup_{j \in \ell} Y_j$, we have $\Delta_{i,e} f(\boldsymbol{y}) = H(\{X_e^i\} \mid \mathcal{S}_{\boldsymbol{y}})$. Since $H(\cdot)$ is monotone, we have $\Delta_{i,e} f(\boldsymbol{y}) \geq 0$.

To see the orthant submodularity, let $\boldsymbol{y} = (Y_1, \ldots, Y_k)$ and $\boldsymbol{y}' = (Y_1', \ldots, Y_k')$ with $\boldsymbol{y} \preceq \boldsymbol{y}'$. Also, let $i \in [k]$ and $e \in V \setminus \bigcup_{j \in [k]} Y_j'$. Then, $\Delta_{e,i} f(\boldsymbol{y}) = H(\{X_e^i\} \mid \mathcal{S}_{\boldsymbol{y}})$ and $\Delta_{e,i} f(\boldsymbol{y}') = H(\{X_e^i\} \mid \mathcal{S}_{\boldsymbol{y}'})$. Since $\mathcal{S}_{\boldsymbol{y}'} \subseteq \mathcal{S}_{\boldsymbol{y}}$, we have $\Delta_{e,i} f(\boldsymbol{y}) \geq \Delta_{e,i} f(\boldsymbol{y}')$ from the submodularity of $H$.