[Reviews · NeurIPS 2015]

Submitted by Assigned_Reviewer_1

*Summary*

This paper studies the constrained monotone k-submodular function maximization problem, where the task is to select k disjoint subsets of items (of bounded total / individual cardinality) that achieve maximal utility. The authors propose (randomized) constant-factor approximation algorithms for such problems which run in almost linear time, and are provably faster than existing algorithms (e.g., greedy submodularity maximization under partition matroid constraints).

*Quality and Clarity*

The paper nicely extends the recent work of [Iwata et al '15] to the constrained maximization setting. The proofs are technically sound, and experimental setup is thoroughly explained. Even though the paper does not provide tightness results of the proposed algorithms, the technical contribution as it is seems to be enough to significantly advance the state of the art.

*Originality and Significance*

As justified in the paper, there has been some very recent work on k-submodular function maximization in the past couple of years. The constrained optimization problem is of significant practical interest, and the proof is insightful for future work.

Randomized algorithms have been used widely to speed up submodular maximization problems e.g., [Buchbinder et al, SODA'14]. It would be nice to see a brief survey in the introduction section.
Summary: Overall, I find this paper theoretically interesting and practically useful. The results nicely generalizes the previously work on maximizing non-negative k-submodular function under the unconstrained setting, and are backed up by thorough experimental results.

Submitted by Assigned_Reviewer_2

It is known that unconstrained maximisation of submodular functions is NP-hard, and, at the same time, admits good approximation algorithms with provable approximation bounds. The paper considers such tasks with two variants of additional size constraints. Two versions of algorithms are considered for each of the respective tasks. A greedy deterministic algorithm and a faster, randomised algorithm involving random subset sampling. Time complexity and approximation bounds are given for each of the discussed algorithms.

The paper is well structured and well written. All presented concepts are comprehensibly explained. The paper seems to be technically correct (I have not checked all proofs given in the supplementary material).

The proposed algorithms are new. However, they appear to be quite similar to the algorithms for the unconstrained version of the problem (as presented in Ward, Zivny, 2014). In my opinion, this somewhat restricts the significance of the presented material.
Summary: The paper proposes and analyses approximation algorithms for maximising monotone k-submodular functions with additional size constraints. In one case the constraint restricts the total size of of the k subsets, whereas in the other one individual constraints are imposed on the size of each of the subsets.

Submitted by Assigned_Reviewer_3

The paper studies k-submodular maximization subject to size constraints. k-submodular functions are generalization of submodular functions and bisubmodular functions. In the classical submodular setting, the objective is to find a set of at most B items so that the valuation of the set is as large as possible. In this setting, the objective is to pick k disjoint sets of at most B items in total so that the valuation of the B items and the particular partition into k subsets is maximized. The algorithms are variants of the classical greedy algorithm: in every step, pick the item with the maximum marginal gain, until the constraint is reached. The analyses also follow closely analyses in previous work but with some technical changes, including applying the sampling idea from [17] to obtain fast algorithms.

The work is a combination of several ideas developed in previous work without much surprise in the analysis. Nonetheless, some proofs do require some technical development, such as the case analysis in theorem 4.1.

The application to sensor placement with multiple sensor types seems reasonable but the influence maximization application seems somewhat artificial, e.g. the constraint of one item per person.

The paper is well-written and easy to read.

A typo on line 160: |supp(o^{(j)}|=B
Summary: The paper studies k-submodular maximization with size constraint, which is a variant of the classical submodular maximization problem. The algorithms and analyses closely follow previous works.

Submitted by Assigned_Reviewer_4

This paper studies the problem of maximizing monotone k-submodular maximization under either a total size constraint or a set of individual size constraints. For both classes of constraints it proposes two variants of greedy algorithms: k-greedy and k-stochastic greedy. For the total size constraint case, both algorithms, in general, provide a guarantee of 1/2. And a guarantee of 1/3 is given for the individual size constraint case. In the experiments, the proposed algorithms are observed to give better function valuation than baseline methods.

This paper study the extension of submodularity to k-submodularity and generalizes the classical greedy algorithm to this setting. However, I think it would very interesting to see more applications of k-submodularity besides sensor placement and influence maximization.

The stochastic variant of the greedy algorithm is very similar to the following paper:

Lazier Than Lazy Greedy, Mirzasoleiman et al AAAI 2015.

Instead of showing that the stochastic greedy gives a solution with bound 1/2 with probability 1-\delta, I think the authors can also give an expectation bound: the stochastic greedy algorithm always gives a solution on expectation with bound 1/2 - \delta.

In the experiments, it seems that the performance of the stochastic greedy sometimes works even better than the greedy algorithm (as shown in Fig 1 and 3), which is different from the results as reported in the lazier than lazy greedy paper. I am hoping the authors could further investigate why and give some comments on this observation.

Minor comments:

For the proposed algorithms, it seems that the authors actually have used the lazy evaluation trick (minoux' trick) in the experiments, but I think it would be better to describe and discuss how to use the lazy evaluation trick when describing the algorithms.

line 60-62: One may not use the computationally expensive algorithm that is tight with bound (1-1/e) for solving submodular maximization under the partition matroid constraint. Instead, a simple greedy algorithm very much the same as the one proposed in this work can solve it with guarantee 1/2.

For the sensor placement experiment, it seems that the objective is defined as the entropy function. But it is unclear how the entropy function is evaluated, since this function requires exponential complexity to evaluate. Please describe in more detail.
Summary: This paper studies the problem of maximizing monotone k-submodular function under either a total size constraint or a set of individual size constraints. For both classes of constraints it proposes two variants of greedy algorithms that admit similar performance guarantee.

Author Feedback
Author rebuttal: We would like to thank all the reviewers for their careful readings and helpful comments. We will address all the minor issues if the paper is accepted. Below, we respond in details.

Reviewer 3:
> I think the authors can also give an expectation bound
As our algorithm outputs 1/2-approximation with probability 1-delta, the expected value obtained by our algorithm is at least (1-delta)*OPT/2 + delta*0 = (1/2-delta/2)OPT. Hence, we think the current statement has a more information.

> The performance of the stochastic greedy sometimes works even better than the greedy algorithm
In our experiment on influence maximization, we simulated the diffusion process 100 times to choose a seed set, which is relatively small, because of the high computation cost. (It might be possible to simulate the diffusion process faster using recent techniques proposed in the influence maximization literature, but we did not try because it is not the scope of this paper and we do not how to adapt them to handle k topics.) Consequently, the approximate value of the influence spread has a relatively high variance, and this might have caused the greedy method to choose seeds with small influence spreads. We suppose that our two methods will produce almost the same results by increasing the number of simulations.

> It would be better to describe and discuss how to use the lazy evaluation trick when describing the algorithms
For k-Greedy-TS (Algorithm 1), by maintaining an upper bound on the gain of inserting each pair (e,i), we can apply the lazy evaluation technique directly. More specifically, we use a priority queue for picking up a pair with the largest (upper bound on the) gain. For k-Stochastic-Greedy-TS (Algorithm 2), we maintain an upper bound on the gain for each pair (e,i), and we pick up a pair in R with the largest (upper bound on the) gain for each iteration. We will describe more how we implemented the lazy evaluation technique if the paper is accepted.

> It is unclear how the entropy function is evaluated
Since Pr[s] log Pr[s] equals 0 when Pr[s] equals 0, we only need to take into account events whose probability is greater than 0. The number of such events is at most the number of given readings, and thus evaluating the entropy function takes linear time in the input size.

Reviewer 4:
> It would be nice to see a brief survey in the introduction section
Thank you for your suggestion. We'll add a survey if the paper is accepted.

Reviewer 6:
The idea of creating sequences of vectors s^0,...,s^B and o^0,...,o^B and analyzing how the function values change in each step is similar to that of [10]. However, in our case, we need to carefully define o^j so that it satisfies the size constraint. This makes the proof more involved. The randomization part is totally new.